# Margin Rectification for Long-Tailed Visual Recognition

## Abstract

Long-tailed visual recognition tasks pose great challenges for neural networks on how to handle the imbalanced predictions between head(common) and tail(rare) classes, i.e., models tend to classify tail classes as head classes. While existing research focused on data resampling and loss function engineering, in this paper, we take a different perspective: the *classification margins*. We study the relationship between the margins and logits and empirically observe that the unrectified margins and logits are *positively correlated*. We propose a simple yet effective *MARgin Rectification* approach (**MARR**) to rectify the margins to obtain better logits. We validate MARR through extensive experiments on common long-tailed benchmarks including CIFAR-LT, ImageNet-LT, Places-LT, and iNaturalist-LT. Experimental results demonstrate that our MARR achieves favorable results on these benchmarks. In addition, MARR is extremely easy to implement with just three lines of code. We hope this simple approach will motivate people to rethink the unrectified margins and logits in long-tailed visual recognition.

## 1 Introduction

Despite the great success of neural networks in the visual recognition field (Simonyan & Zisserman, 2014; He et al., 2016), it is still challenging for neural networks to deal with the ubiquitous long-tailed datasets in the real world (Buda et al., 2018; Kang et al., 2019; Zhou et al., 2020). To be clear, in the long-tailed datasets, the high-frequency classes (head/common classes) occupy most of the instances, whereas the low-frequency classes (tail/rare classes) involve a small amount of instances (Liu et al., 2019; Van Horn & Perona, 2017). Due to the imbalance of the training data, the model performs well in head classes and its performance is much worse in tail classes (Buda et al., 2018; Zhang et al., 2021).

Towards addressing the long-tailed recognition problem, there are several strategies such as data re-sampling and loss function engineering. Data re-sampling aims to 'simulate' a balanced training dataset by over-sampling the tail class or under-sampling the head classes (Ando & Huang, 2017; Buda et al., 2018; Pouyanfar et al., 2018; Shen et al., 2016), while loss re-weighting is introduced to adjust the weights of losses for different classes or different instances (Byrd & Lipton, 2019; Khan et al., 2017; Wang et al., 2017). For more balanced gradients between classes, some class-balanced loss functions adjust the logits instead of weighting the losses (Menon et al., 2020; Cao et al., 2019; Ren et al., 2020).

However, as pointed out by existing research (Ganganwar, 2012; Zhou & Liu, 2005; Cao et al., 2019), data re-sampling strategies and loss re-weighting schemes will possibly cause underfitting on the head class and overfitting on the tail class. On the other hand, class-balanced loss functions or data re-sampling will lead to worse data representations compared with the standard training using the cross-entropy loss and the instance-balanced sampling (i.e., each instance has the same probability of being sampled) (Kang et al., 2019; Ren et al., 2020). In addition, recent research reveals that the unrectified decision boundary given by the classifier head seems to be the performance bottleneck of the long-tailed visual recognition (Kang et al., 2019; Zhang et al., 2021). To benefit from both good data representations and the unbiased decision boundary, Decoupling, a heuristic two-stage strategy is proposed to adjust the initially-learned classifier head (Kang et al., 2019) after the standard training. Furthermore, distribution alignment (Zhang et al., 2021) is developed as an adaptive calibration function to adjust the initially trained logits for each data point. However, as pointed out by existing research (Platt et al., 1999; Elsayed et al., 2018), the margins

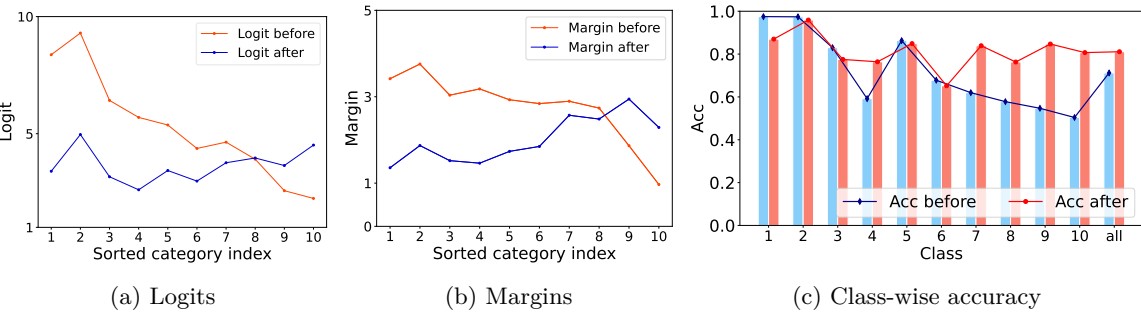

(a) Logits          (b) Margins          (c) Class-wise accuracy

Figure 1: **Logits, margins, and class-wise accuracy of CIFAR-10-LT with imbalance factor 200**. Here, the logit and margin represent the average logit and margin of each class, *before* and *after* refers to standard training and our method, respectively. The class indices are sorted by the number of samples (Head to tail).

and logits have a critical effect to the classification performance. *The relation between the unrectified margin and the logits is neglected in existing research*, where the margin is the distance from the data point to the decision boundary.

In this paper, we study the relationship between margins and logits, which are critical factors that dominate the long-tailed performance. As shown in Figure 1, we empirically find that the margin and the logit are correlated with the cardinality of each class. To be concrete, before any rectification, *head classes tend to have much larger margins and logits than tail classes*. Therefore, it is necessary to rectify the margin to obtain the balanced logits. More importantly, as shown in Figure 1c, the unrectified margins and logits will have a negative impact on the classification performance. Therefore, it remains challenging to design an efficient method for such rectification that can achieve satisfying performance without introducing much computational burden.

Inspired by the above phenomenon, we propose a simple yet effective **MAR**gin **R**ectification (**MARR**) approach for long-tailed recognition. In detail, after getting the representations and the classifier head from the standard training, we propose a simple class-specific margin rectification function with only $2K$ learnable parameters to adjust the initially learned margins, where $K$ is the number of classes. As demonstrated in Figure 1, the logits are more balanced when using MARR. We conduct experiments on several popular long-tail benchmark datasets: CIFAR-10-LT (Krizhevsky et al., 2009), CIFAR-100-LT (Krizhevsky et al., 2009), Places-LT (Zhou et al., 2017), iNaturalist 2018 (Van Horn et al., 2018), and ImageNet-LT (Liu et al., 2019). The results demonstrate that our proposed MARR approach performs remarkably well while remaining very simple to implement. We hope that our exploration will attract attention to the imbalanced margins in long-tailed recognition. To sum up, our contributions are as follows:

- For the first time in long-tailed recognition, we study the unrectified predictions from a margin-based perspective. We empirically find that unrectified margins will cause imperfect predictions, which could lead to future algorithm designs.

- Based on our observations, we propose a simple yet effective margin rectification (MARR) approach with only $2K$ trainable parameters to adjust the margins to get the unbiased prediction for long-tailed visual recognition problem.

- Compared with SOTA methods Zhang et al. (2021); Hong et al. (2021), MARR is competitive on various long-tailed visual benchmarks like CIFAR-10-LT (Krizhevsky et al., 2009), CIFAR-100-LT (Krizhevsky et al., 2009), ImageNet-LT (Liu et al., 2019), Places-LT (Zhou et al., 2017) and iNaturalist2018 (Van Horn et al., 2018). In addition, it is extremely easy to implement with just three lines of code.

## 2    Related Work

Long-tailed visual recognition has attracted much attention for its commonness in the real world (He & Garcia, 2009; Buda et al., 2018; Kang et al., 2019; Ren et al., 2020; Yang & Xu, 2020; Hong et al., 2021). Existing methods can be divided into four categories.

**Data re-sampling.** Data re-sampling techniques re-sample the imbalanced training dataset to 'simulate' a balanced training dataset. These methods include under-sampling, over-sampling, and classed-balanced sampling. Under-sampling decreases the probability of the instance of head classes being sampled (Drummond et al., 2003), whereas over-sampling makes instances of tail classes more likely to be sampled (Chawla et al., 2002; Han et al., 2005; Wang et al., 2021a). Class-aware sampling chooses instances of each class with the same probabilities (Shen et al., 2016).

**Loss function engineering.** Loss function engineering is another direction to obtain balanced gradients during the training. The typical methods can be categorized as loss-reweighting and logits adjustment. Loss re-weighting adjusts the weights of losses for different classes or different instances in a more balanced manner, i.e. the instances in tail classes have larger weights than those in head classes (Byrd & Lipton, 2019; Khan et al., 2017; Wang et al., 2017). On the other hand, instead of re-weight losses, some class-balanced loss functions adjust the logits to get balanced gradients during training (Menon et al., 2020; Cao et al., 2019; Ren et al., 2020; Yang et al., 2009).

**Decision boundary adjustment.** Nevertheless, data re-sampling or loss function engineering will influence the representations of data (Ren et al., 2020). Lots of empirical observations show that we can acquire good representation when using the standard training and the classifier head is the performance bottleneck (Kang et al., 2019; Zhang et al., 2021; Yu et al., 2020; Kim & Kim, 2020). To solve the above problem, decision boundary methods re-adjust the classifier head after the standard training in a learnable way (Kang et al., 2019; Zhang et al., 2021) or using maximum likelihood estimation such as the Platt scaling Platt et al. (1999). However, they ignore the relationship between the unrectified margins and logits. Moreover, MARR targets a multi-class classification problem and Platting scaling targets binary classification (based on sigmoid), and their updating manners are also different since MARR adopts an end-to-end manner that updates $2K$ learnable parameters.

**Other methods.** There also exist other paradigms to deal with the long-tailed recognition task, including task-specific architecture design (Wang et al., 2021c; Zhou et al., 2020; Wang et al., 2021a), transfer learning (Liu et al., 2019; Yin et al., 2019), domain adaptation (Jamal et al., 2020), semi supervised learning and self supervised learning (Yang & Xu, 2020). But these methods either rely on the non-trivial architecture design or external data. In contrast, our proposed MARR is very simple to implement and does not require external data. The detailed comparison between MARR and similar methods is shown in Table 1.

## 3    Method

### 3.1    Preliminaries

In the popular setting of long-tailed recognition (Kang et al., 2019; Cui et al., 2019; Ren et al., 2020), the training data distribution is imbalanced while the test data distribution is balanced. More formally, let $\mathcal{D} = \{(\mathbf{x}_i, y_i)\}_{i=1}^n$ be a training set, where $y_i$ denotes the label of data point $\mathbf{x}_i$. Specifically, $n = \sum_{j=1}^K n_j$ is the total number of training samples, where $n_j$ is the number of training samples in class $j$ and $K$ is the number of classes. We assume $n_1 > n_2 > \cdots > n_K$ without loss of generality. Normally, the prediction function is composed of two modules: the feature representation learning function $f : \mathbf{x} \mapsto \mathbf{z}$ parameterized by $\theta_r$ and the classifier $g : \mathbf{z} \mapsto y$ parameterized by $\theta_c$, where $\mathbf{z} \in \mathbb{R}^p$ denotes the feature representation and $p$ is the feature dimension. Typically, $g$ is a linear classifier that gives the classification score of class $j$ as:

$$\eta_j = g(\mathbf{z}) := \mathbf{W}_j \mathbf{z} + \mathbf{b}_j, \tag{1}$$

where $\mathbf{W}_j$ and $\mathbf{b}_j$ are the weight vector and bias for class $j$, respectively. Finally, using the softmax function, the probability of $\mathbf{x}_i$ being classified as label $y_i$ is expressed as:

$$p(y = y_i|\mathbf{x}_i; \theta_r, \theta_c) = \frac{\exp(\eta_{y_i})}{\sum_{j=1}^{K} \exp(\eta_j)}, \tag{2}$$

and its loss is computed as the cross-entropy loss:

$$\ell(\mathbf{x}_i, y_i; \theta_r, \theta_c) = -\log\left(\frac{\exp(\eta_{y_i})}{\sum_{j=1}^{K} \exp(\eta_j)}\right). \tag{3}$$

## 3.2 Unrectified Margins and Logits

The decision boundary is often unrectified in long-tailed recognition, which will lead to imperfect predictions, i.e., the model tends to classify tail classes as head classes. To alleviate this issue, data re-sampling and loss function engineering are two directions to simulate a 'balanced' training dataset. However, such techniques will do harm to the representation learning of the model and lead to low overall accuracy (Kang et al., 2019; Ren et al., 2020). These methods including ours will sometimes do harm to the performance of head classes. However, the "good" performance of head classes is sometimes not always ensuring positive overall results. To benefit from both the good representation that will improve the overall accuracy and the rectified decision boundary, decision boundary adjustment methods are developed (Kang et al., 2019; Zhang et al., 2021). However, existing decision boundary adjustment methods ignore

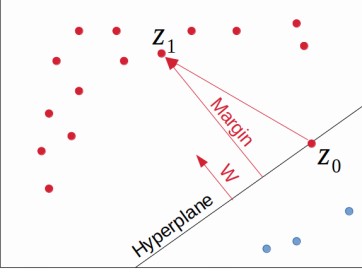

Figure 2: **Illustration of margins.** The red and blue dots denote majority and minority classes, respectively.

such rectification in the margins, which is essential to avoid unrectified predictions. Thus, we aim to rectify the margins to obtain balanced predictions.

In this paper, we find that the margins Hastie et al. (2009) and logits are biased in long-tailed recognition. The margins are illustrated in Figure 2. We define an affine hyperplane $H_j \in \mathbb{R}^{p-1}$ of class $j$ as $\mathbf{W}_j\mathbf{z} + \mathbf{b}_j = 0$, i.e. any representation point falling on the positive side of $H_j$ can be attributed to class $j$. Assume that $\mathbf{z}_0$ is a point satisfying $\mathbf{W}_j\mathbf{z}_0 + \mathbf{b}_j = 0$, i.e., $\mathbf{z}_0$ is on the hyperplane $H_j$. Suppose $\mathbf{z}_1$ is an arbitrary point in the feature space. We construct the vector $\mathbf{z}_1 - \mathbf{z}_0$ pointing from $\mathbf{z}_0$ to $\mathbf{z}_1$ and project it onto the normal vector $\mathbf{W}_j$. The length of the projection vector $proj_{\mathbf{W}_j}(\mathbf{z}_1 - \mathbf{z}_0)$ is the margin from $\mathbf{z}_1$ to $H_j$. More formally, such margin is calculated as:

$$\begin{aligned}
d_j &= \left\| proj_{\mathbf{W}_j}(\mathbf{z}_1 - \mathbf{z}_0) \right\| \\
&= \left\| \frac{\mathbf{W}_j \cdot (\mathbf{z}_1 - \mathbf{z}_0)}{\mathbf{W}_j \cdot \mathbf{W}_j} \mathbf{W}_j \right\| \\
&= \frac{\mathbf{W}_j \cdot \mathbf{z}_1 - \mathbf{W}_j \cdot \mathbf{z}_0}{\|\mathbf{W}_j\|} \\
&= \frac{\mathbf{W}_j\mathbf{z}_1 + \mathbf{b}_j}{\|\mathbf{W}_j\|} \quad (\text{since } \mathbf{W}_j\mathbf{z}_0 + \mathbf{b}_j = 0),
\end{aligned} \tag{4}$$

where $\|\cdot\|$ denotes L2 norm. Thus, the logit $\mathbf{W}_j \cdot \mathbf{z}_1 + \mathbf{b}_j$ can also be expressed as $\|\mathbf{W}_j\|d_j$. Based on this conclusion, we can rewrite equation 2 as:

$$p(y = y_i|\mathbf{x}_i; \theta_r, \theta_c) = \frac{\exp(\eta_{y_i})}{\sum_{j=1}^{K} \exp(\eta_j)} = \frac{\exp(\|\mathbf{W}_{y_i}\|d_{y_i})}{\sum_{j=1}^{K} \exp(\|\mathbf{W}_j\|d_j)}. \tag{5}$$

Consider a data point is on the decision boundary of class $j$ and class $t$ (on the hyperplane in Figure 2), i.e., such data point has the same probability of being classified as class $j$ or class $t$. Clearly, the assumed data point on the decision boundary satisfies:

$$\eta_j = \eta_t = \|\mathbf{W}_j\|d_j = \|\mathbf{W}_t\|d_t. \tag{6}$$

According to equation 6, data will be classified as class $t$ because $d_j < d_t$ when $\|\mathbf{W}_j\| = \|\mathbf{W}_t\|$. And our empirical observations show that head classes tend to have much larger margins and logits than tail classes:

$$\begin{aligned}
&\bar{d}_1 > \bar{d}_2 > \cdots > \bar{d}_K, \\
&\bar{\eta}_1 > \bar{\eta}_2 > \cdots > \bar{\eta}_K, \\
&\text{if} \quad n_1 > n_2 > \cdots > n_K,
\end{aligned} \tag{7}$$

where $\bar{d}_j$ is the average margin of class $j$ and $\bar{\eta}_j$ is the average logit of class $j$ after the standard training. In detail, on the sub-dataset $\mathcal{D}_j = \{(\mathbf{x}_i, y_i = j)\}_{i=1}^{n_j}$, $\bar{\eta}_j = \frac{1}{n_j}\eta_j$, $\bar{d}_j = \frac{\bar{\eta}_j}{\|\mathbf{W}_j\|}$.

### 3.3   Margin Rectification (MARR)

To get the rectified logits, we propose MARR to rectify the margins after the standard training. Concretely speaking, we train a simple class-specific margin rectification model with the original margin fixed:

$$\hat{d}_j = \omega_j \cdot d_j + \beta_j, \tag{8}$$

where $\omega_j$ and $\beta_j$ are learnable parameters for class $j$ and $j \in [1, K]$. It is worth noting that MARR is extended from class $j$ to the whole dataset in practise. In other words, MARR only has $2K$ trainable parameters. Thus, the rectified logit is computed as:

$$\begin{aligned}
\|\mathbf{W}_j\|\hat{d}_j &= \|\mathbf{W}_j\|(\omega_j \cdot d_j + \beta_j) \\
&= \omega_j \cdot \|\mathbf{W}_j\|d_j + \beta_j \cdot \|\mathbf{W}_j\| \\
&= \omega_j \cdot \eta_j + \beta_j \cdot \|\mathbf{W}_j\|,
\end{aligned} \tag{9}$$

where $\eta_j$ is the initial fixed logit. Then, we can get the rectified prediction distribution:

$$p(y = y_i|\mathbf{x}_i; \theta_r, \theta_c) = \frac{\exp(\omega_{y_i} \cdot \eta_{y_i} + \beta_{y_i} \cdot \|\mathbf{W}_{y_i}\|)}{\sum_{j=1}^{K} \exp(\omega_j \cdot \eta_j + \beta_j \cdot \|\mathbf{W}_j\|)}. \tag{10}$$

The training process of the margin rectification approach can be written with *just three* lines of Pytorch codes as shown in Line 4-6 of Algorithm 1.

---

**Algorithm 1** The torch-like code for MARR.

---

1: Initialization of the margin rectification approach:
   ```
   omega=torch.nn.Parameter(torch.ones(1,K))
   beta=torch.nn.Parameter(torch.zeros(1,K))
   ```
2: **Input:** training data x, standard pre-trained neural network model.
3: ```with torch.no_grad():```
4: ```    w_norm = torch.norm(model.fc.weight, dim=1)```
5: ```    logit_before = model(x)```
6: ```logit_after = omega * logit_before + beta * w_norm```
7: Compute loss and update parameters of omega and beta.

---

Furthermore, for more balanced gradients during training, we re-weight the loss as the previous work does (Zhang et al., 2021). Finally, the loss for training the margin rectification approach is:

$$\ell(\mathbf{x}_i, y_i; \tilde{\theta}_r, \tilde{\theta}_c, \omega, \beta) = -U_{y_i} \cdot \log\left(\frac{\exp\left(\omega_{y_i} \cdot \eta_{y_i} + \beta_{y_i} \cdot \|\mathbf{W}_{y_i}\|\right)}{\sum_{j=1}^{K} \exp(\omega_j \cdot \eta_j + \beta_j \cdot \|\mathbf{W}_j\|)}\right), \tag{11}$$

where $\tilde{\theta}_r$ and $\tilde{\theta}_c$ denote that these parameters are frozen during training. The weight for class $y_i$ is calculated as:

$$U_{y_i} = K \cdot \frac{(1/n_{y_i})^\gamma}{\sum_{j=1}^{K}(1/n_j)^\gamma}, \tag{12}$$

---

**Algorithm 2** The detailed training procedure including both standard training and margin rectification function training.

---

1: **Input:** The training dataset $\mathcal{D} = \{(\mathbf{x}_i, y_i)\}_{i=1}^n$, the parameters of the representation function $\theta_r$, the parameters of the classifier $\theta_c$, the parameters of the margin rectification function $\omega$ and $\beta$, the number of classes $K$ and the pre-defined scale hyper-parameter $\gamma$.

2: First stage: the standard training use the instance-balanced sampling and the cross entropy loss.

3: **while** not reach the maximum iteration **do**

4:   Use instance-balanced sampling to sample a batch of data $\mathcal{D}_s = \{(\mathbf{x}_i, y_i)\}_{i=1}^s$ from the training dataset $\mathcal{D}$, where $s$ is the batch size.

5:   Compute the loss and update the model parameters.

$\ell(\mathcal{D}_s; \theta_r, \theta_c) = \frac{-1}{s} \sum_{i=1}^s \log\left(\frac{\exp(\eta_{y_i})}{\sum_{j=1}^K \exp(\eta_j)}\right)$, where $\eta_j$ is the classification score of class $j$.

6: **end while**

7: Second stage: Rectify the margins trained in the first stage.

8: **while** not reach the maximum iteration **do**

9:   Use instance-balanced sampling to sample a batch of data $\mathcal{D}_s = \{(\mathbf{x}_i, y_i)\}_{i=1}^s$ from the training dataset $\mathcal{D}$, where $s$ is the batch size.

10:   Compute the loss and update the model parameters.

$\ell(\mathcal{D}_s; \tilde{\theta}_r, \tilde{\theta}_c, \omega, \beta) = \frac{1}{s} \sum_{i=1}^s (-U_{y_i} \cdot \log\left(\frac{\exp(\omega_{y_i} \cdot \eta_{y_i} + \beta_{y_i} \cdot \|\mathbf{W}_{y_i}\|)}{\sum_{j=1}^K \exp(\omega_j \cdot \eta_j + \beta_j \cdot \|\mathbf{W}_j\|)}\right)$, where parameters with $\tilde{\ }$ are fixed during training and $U_j$ is calculated as shown in Eq. 12.

11: **end while**

12: **Return:** Model parameters $\theta_r, \theta_c, \omega, \beta$.

---

where $\gamma$ is a scale hyper-parameter. When $\gamma = 0$, the weight for all classes is 1, which means no re-weighting at all.

To be more clear, the whole detailed training procedure including both standard training and margin rectification function training is demonstrated in Algorithm 2. Lines 2-6 include the training procedure of the standard training using the instance-balanced sampling and the cross-entropy loss. Lines 7-11 contain the training process of our margin rectification function. It is worth noting that in the second stage, parameters $\theta_r$ and $\theta_c$ are all fixed.

### 3.4 Discussion

We clarify the differences between MARR and other learnable decision boundary adjustment methods in detail. As shown in Table 1, Decouple-cRT (Kang et al., 2019) retrains the whole parameters of the classifier, while Decouple-LWS (Kang et al., 2019) only adjusts the norm of weight vectors $\|\mathbf{W}_j\|$. Instead of adjusting the classifier head, DisAlign (Zhang et al., 2021) chooses to rectify the logit for each data point. But their rectification method is heuristic that simply adds the rectified logit and the original logit with a re-weighting scheme. To be more clear, the weighted sum of logits for DisAlign is $\sigma(\mathbf{z}_j)(\omega_j \eta_j + \beta) + (1 - \sigma(\mathbf{z}_j))\eta_j$, where $\sigma(\cdot)$ is an instance-specific confidence function. However, different from previous methods, our MARR focuses on rectifying the margin which we believe is the performance bottleneck of the long-tailed classifier.

Table 1: The difference between MARR and other decision boundary adjustment methods. $j \in [1, K]$ is class index.

| Method | Rectification method |
|---|---|
| Decouple-cRT (Kang et al., 2019) | retrain $\mathbf{W}_j, \mathbf{b}_j$ |
| Decouple-LWS (Kang et al., 2019) | $\|\mathbf{W}_j\|^{1-\omega_j}$ |
| DisAlign (Zhang et al., 2021) | $\sigma(\mathbf{z}_j)(\omega_j \eta_j + \beta) + (1 - \sigma(\mathbf{z}_j))\eta_j$ |
| MARR | $\omega_j \cdot d_j + \beta_j$ |

# 4 Experiments

In this section, we conduct extensive experiments compared with the state-of-the-art methods to validate the effectiveness of MARR. Firstly, we report the performance on common benchmarks like CIFAR-10-LT (Krizhevsky et al., 2009), CIFAR-100-LT (Krizhevsky et al., 2009), ImageNet-LT (Liu et al., 2019), Places-LT (Zhou et al., 2017) and iNaturalist2018 (Van Horn et al., 2018). The results of MARR are competitive even though MARR is simple. Then we conduct further analysis to explain the reason for the success of MARR.

## 4.1 Setup

**Datasets**   We follow the common evaluation protocol (Liu et al., 2019) and conduct experiments on CIFAR-10-LT (Krizhevsky et al., 2009), CIFAR-100-LT (Krizhevsky et al., 2009), ImageNet-LT (Liu et al., 2019), Places-LT (Zhou et al., 2017) and iNaturalist2018 (Van Horn et al., 2018). The imbalance factor used in CIFAR datasets is defined as $N_{max}/N_{min}$ where $N_{max}$ is the number of samples on the largest class and $N_{min}$ the smallest. We report CIFAR results with two different imbalance ratios: 100 and 200. For ImageNet-LT and Places-LT experiments, we further split classes into three sets: Many-shot (with more than 100 images), Medium-shot (with 20 to 100 images), and Few-shot (with less than 20 images).

**Training Configuration**   For a fair comparison, our experiments are conducted under the most commonly used codebase of long-tailed studies: Open Long-Tailed Recognition (OLTR) (Liu et al., 2019), using Py-Torch (Paszke et al., 2019) framework. The model structures used for CIFAR, ImageNet-LT, Places-LT and iNaturalist18 datasets are ResNet32, ResNeXt50, ResNet152 and ResNet50, respectively. The model for Places-LT is pre-trained on the full ImageNet-2012 dataset while models for other datasets are trained from scratch. For ImageNet-LT, Places-LT, and iNaturalist18, we train 90, 30, and 200 epochs in the first standard training stage; and 10, 10, and 30 epochs in the second margin rectification stage, with the batch size of 256, 128, and 256, respectively. For CIFAR-10-LT and CIFAR-100-LT, the models are trained for 13,000 iterations with a batch size of 512. We use the SGD optimizer with momentum 0.9 and weight decay $5e - 4$

Table 2: Accuracy on CIFAR-10-LT and CIFAR-100-LT datasets with different imbalance ratios.

| Dataset | CIFAR-10-LT | | CIFAR-100-LT | |
|---|---|---|---|---|
| Imbalance Factor | 100 | 200 | 100 | 200 |
| Softmax | 78.7 | 74.4 | 45.3 | 41.0 |
| Data Re-sampling | | | | |
| Class Balanced Sampling (CBS) | 77.8 | 68.3 | 42.6 | 37.8 |
| Loss Function Engineering | | | | |
| Class Balanced Weighting (CBW) | 78.6 | 72.5 | 42.3 | 36.7 |
| Class Balanced Loss (Cui et al., 2019) | 78.2 | 72.6 | 44.6 | 39.9 |
| Focal Loss (Lin et al., 2017) | 77.1 | 71.8 | 43.8 | 40.2 |
| LADE (Hong et al., 2021) | 81.8 | 76.9 | 45.4 | 43.6 |
| LDAM (Cao et al., 2019) | 78.9 | 73.6 | 46.1 | 41.3 |
| Equalization Loss (Tan et al., 2020) | 78.5 | 74.6 | 47.4 | 43.3 |
| Balanced Softmax (Ren et al., 2020) | 83.1 | 79.0 | 50.3 | 45.9 |
| Decision Boundary Adjustment | | | | |
| DisAlign (Zhang et al., 2021) | 78.0 | 71.2 | 49.1 | 43.6 |
| Decouple-cRT (Kang et al., 2019) | 82.0 | 76.6 | 50.0 | 44.5 |
| Decouple-LWS (Kang et al., 2019) | 83.7 | 78.1 | 50.5 | 45.3 |
| Others | | | | |
| BBN (Zhou et al., 2020) | 79.8 | - | 42.6 | - |
| Hybrid-SC (Wang et al., 2021c) | 81.4 | - | 46.7 | - |
| MARR | **85.3** | **81.1** | **50.8** | **47.4** |

Table 3: The performance on ImageNet-LT.

| Method | Many | Medium | Few | Overall |
|---|---|---|---|---|
| Softmax | 65.1 | 35.7 | 6.6 | 43.1 |
| Loss Function Engineering | | | | |
| Focal Loss (Lin et al., 2017) | 64.3 | 37.1 | 8.2 | 43.7 |
| Seesaw (Wang et al., 2021b) | **67.1** | 45.2 | 21.4 | 50.4 |
| Balanced Softmax (Ren et al., 2020) | 62.2 | 48.8 | 29.8 | 51.4 |
| LADE (Hong et al., 2021) | 62.3 | 49.3 | 31.2 | 51.9 |
| Decision Boundary Adjustment | | | | |
| Decouple-$\pi$-norm (Kang et al., 2019) | 59.1 | 46.9 | 30.7 | 49.4 |
| Decouple-cRT (Kang et al., 2019) | 61.8 | 46.2 | 27.4 | 49.6 |
| Decouple-LWS (Kang et al., 2019) | 60.2 | 47.2 | 30.3 | 49.9 |
| DisAlign (Zhang et al., 2021) | 60.8 | **50.4** | 34.7 | 52.2 |
| Others | | | | |
| OLTR (Liu et al., 2019) | 51.0 | 40.8 | 20.8 | 41.9 |
| Causal Norm (Tang et al., 2020) | 62.7 | 48.8 | 31.6 | 51.8 |
| MARR | 60.4 | 50.3 | **36.6** | **52.3** |

for all datasets except for iNaturalist18 where the weight decay is $1e-4$. In the standard training stage, we use a cosine learning rate schedule with an initial value of 0.05 for CIFAR and 0.1 for other datasets, which gradually decays to 0. In the margin rectification stage, we use a cosine learning rate schedule with an initial learning rate starting from 0.05 to 0 for all datasets. $\gamma$ is set to 1.2 for all datasets. The hyper-parameters of compared methods follow their paper. *For fairness, we use the same pre-trained model for decision boundary adjustment methods.*

## 4.2 Comparison with previous methods

In this section, we compare the performance of MARR to other recent works. We select some recent methods from each of the following four categories for comparison: data re-sampling, loss function engineering, decision boundary adjustment, and others. The standard training with the cross-entropy loss and instance balance sampling is called Softmax in our results.

**CIFAR** Table 2 presents results for CIFAR-10-LT and CIFAR-100-LT. MARR outperforms all other methods in CIFAR-LT. Compared with other decision boundary adjustment methods, MARR shows favorable results. The accuracy of MARR outruns Decouple-LWS 1.6%, 3%, 0.3% and 1.9% on CIFAR-10-LT(100), CIFAR-10-LT(200), CIFAR-100-LT(100) and CIFAR-100-LT(200) respectively, where (·) denotes the imbalance factor. In addition, MARR outperforms all data re-sampling and loss function engineering methods that my need laborious hyper-parameter. The performance of well-designed networks such as BBN and Hybrid-SC are also not as good as that of MARR.

**ImageNet-LT** We further evaluate MARR on the ImageNet-LT dataset. As Table 3 shows, MARR is better than all loss function engineering methods. Compared with LADE, although our overall accuracy is 0.4% higher, the accuracy on the few-shot classes is 5.4% higher. The few-shot accuracy and overall accuracy of MARR are 1.9% and 0.1% higher than DisAlign respectively. Our results are quite surprising considering the simplicity of MARR (See next section for time comparison).

**Places-LT** For the Places-LT dataset, MARR achieves better performance than other decision boundary adjustment methods. Though our overall accuracy is lower than LADE, our few-shot accuracy is still 1.4% higher than LADE. Though MARR is not the best in Places-LT, the results of MARR are still competitive compared with other methods.

Table 4: The performances on Places-LT, starting from an ImageNet pre-trained ResNet-152.

| Method | Many | Medium | Few | Overall |
|---|---|---|---|---|
| Softmax | 46.4 | 27.9 | 12.5 | 31.5 |
| Loss Function Engineering | | | | |
| Focal Loss (Lin et al., 2017) | 41.1 | 34.8 | 22.4 | 34.6 |
| Balanced Softmax (Ren et al., 2020) | 42.0 | 38.0 | 17.2 | 35.4 |
| LADE (Hong et al., 2021) | 42.8 | 39.0 | 31.2 | **38.8** |
| Decision Boundary Adjustment | | | | |
| Decouple-LWS (Kang et al., 2019) | 40.6 | 39.1 | 28.6 | 37.6 |
| Decouple-$\pi$-norm (Kang et al., 2019) | 37.8 | **40.7** | 31.8 | 37.9 |
| DisAlign (Zhang et al., 2021) | 40.0 | 39.6 | 32.3 | 38.3 |
| Others | | | | |
| OLTR (Liu et al., 2019) | **44.7** | 37.0 | 25.3 | 35.9 |
| Causal Norm (Tang et al., 2020) | 23.8 | 35.8 | **40.4** | 32.4 |
| MARR | 39.9 | 39.8 | 32.6 | 38.4 |

**iNaturalist-LT**   Finally, we present the Top-1 accuracy results for iNaturalist-LT dataset in Table 5. We can observe a similar trend that our proposed method wins all the existing approaches and surpasses DisAlign by 0.1% absolute improvement.

Table 5: Top-1 accuracy on iNaturalist-LT.

| Method | Top-1 Accuracy |
|---|---|
| Softmax | 65.0 |
| Loss Function Engineering | |
| Class Balanced Loss (Cui et al., 2019) | 61.1 |
| LDAM (Cao et al., 2019) | 64.6 |
| Balanced Softmax (Ren et al., 2020) | 69.8 |
| LADE (Hong et al., 2021) | 70.0 |
| Decision Boundary Adjustment | |
| Decouple-$\pi$-norm (Kang et al., 2019) | 69.3 |
| Decouple-LWS (Kang et al., 2019) | 69.5 |
| DisAlign (Zhang et al., 2021) | 70.3 |
| Others | |
| Casual Norm (Tang et al., 2020) | 63.9 |
| Hybrid-SC (Wang et al., 2021c) | 68.1 |
| MARR | **70.4** |

## 4.3 Effectiveness validation

**Comparison of the trainable parameters of decision boundary adjustment methods**   As shown in Table 6, MARR achieves the best performance among other compared methods on CIFAR-100-LT(200) and ImageNet-LT. Even though our trainable parameters are more than Decouple-LWS, our performance is better. Besides, it is surprising that MARR obtains such a favorable performance with only so few parameters.

Table 6: **Comparison of the trainable parameters (#Param.) of the learnable decision boundary adjustment methods.** $p$ is the feature dimension and $K$ is the number of classes (ResNeXt50 for ImageNet-LT: $p = 2048, K = 1000$).

| Method | CIFAR-100-LT | ImageNet-LT | #Param. |
|---|---|---|---|
| Decouple-cRT | 44.5 | 49.6 | $pK + K$ |
| Decouple-LWS | 45.3 | 49.9 | $K$ |
| DisAlign | 43.6 | 52.2 | $p + 2K$ |
| MARR | **47.4** | **52.3** | $2K$ |

**Effects of different standard pre-trained model** We use different standard pre-trained models on CIFAR-100-LT(200) to explore their effects. As Table 7 illustrates, as the pre-trained dataset gets more balanced, the performance of our margin rectification method gets better. This shows that when comparing decision boundary methods, using the same standard training model is very important for fairness. And the codebase will also affect the standard training. *This is the reason why the result of DisAlign in our paper is inconsistent with the original paper since we cannot get the same standard pre-trained model they used.* So instead, we use our own standard pre-trained model for MARR and DisAlign for fairness. Table 7 also demonstrates that the margin rectification method can achieve better performance when given better representations. So our future works include how to get better representations.

Table 7: Top-1 accuracy on CIFAR-100-LT(200) with different standard pre-trained models.

| Standard pre-trained dataset | Top-1 Accuracy |
|---|---|
| CIFAR-100-LT(200) | 47.1 |
| CIFAR-100-LT(100) | 50.7 |
| CIFAR-100-LT(50) | 54.5 |

## 4.4 Further analysis

In this section, we conduct different experiments for further analysis. To be concrete, we empirically show that MARR can achieve more balanced margins and logits compared with DisAlign. Moreover, the class-wise accuracy of MARR is much better than the standard training baseline model on CIFAR-LT, which indicates that we can alleviate the imbalanced prediction problem and reduce the performance gap between the head classes and the tail classes.

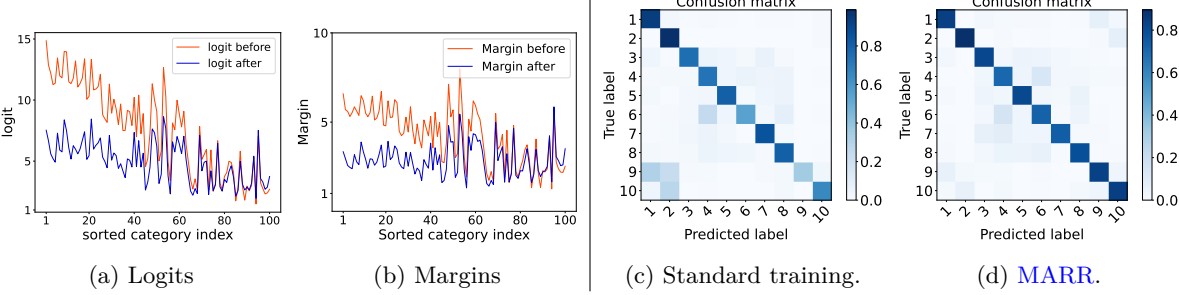

(a) Logits      (b) Margins      (c) Standard training.      (d) MARR.

Figure 3: (a) and (b): **Logits and margins of CIFAR-100-LT(200)**. Here, the logit and margin represent the average logit and margin of each class, *before* refers to the standard training results and *after* refers to the results after our rectification method. The class indices are sorted by the number of samples (Head to tail). (c) and (d): **Confusion matrix of the standard training and MARR on long-tailed CIFAR-10-LT(200).** The fading color of diagonal elements refers to the disparity of the accuracy.

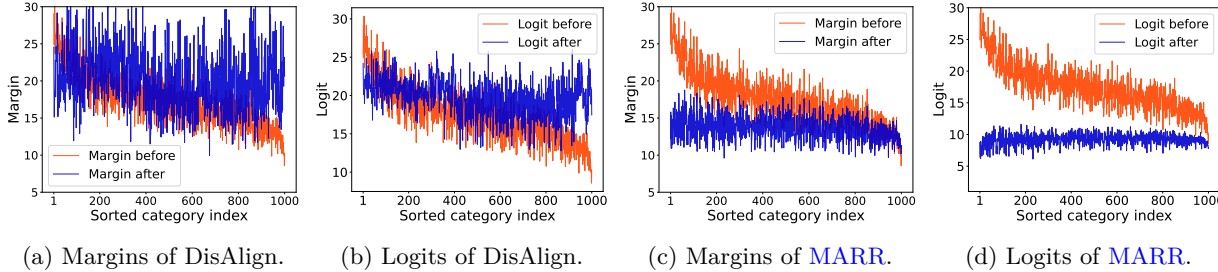

(a) Margins of DisAlign.  (b) Logits of DisAlign.  (c) Margins of MARR.  (d) Logits of MARR.

Figure 5: **The values of margins and logits for each class on the ImageNet-LT dataset.** The logit and margin represent the average logit and margin of each class. The class indices are sorted by the number of samples (Head to tail). *Before* refers to the standard training results and *after* refers to the results after using the decision boundary adjustment method.

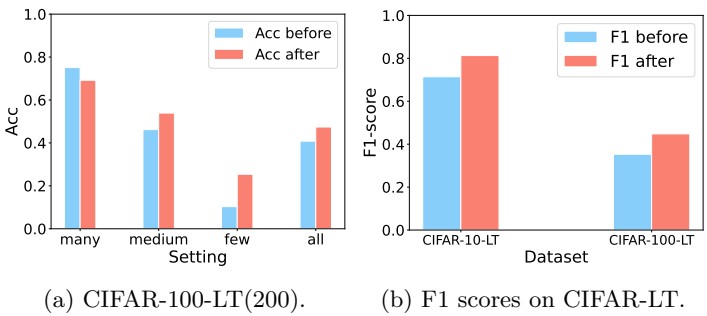

(a) CIFAR-100-LT(200).  (b) F1 scores on CIFAR-LT.

Figure 6: The detailed performance of CIFAR-100-LT(200). *Before* refers to the standard training results and *after* refers to the results after MARR.

**Visualization of the margin and logit**   In this subsection, we visualize the values of margins and logits for each class to show the effect of MARR. As illustrated both in Figure 3, Figure 5c and Figure 5d, before margin rectification, the margins and the logits are unrectified, i.e. the head classes tend to have much larger margins and logits than tail classes. We believe the bias in margins and logits will lead to imperfect predictions in long-tailed visual recognition. The margins and logits become more balanced after the margin rectification. This result proves that we can get better predictions by calibrating the margin. Moreover, as shown in Figure 5a and Figure 5b, MARR will obtain more balanced margins and gradients than DisAlign. The instability of DisAlign may be caused by their heuristic design of the combination of the rectified logits and the origin logits.

**Class-wise performance on CIFAR-LT**   As we can see in Figure 3c and 3d, after our margin rectification method, the performance on tail classes is improved while that on head classes is not severely affected. More intuitively, Figure 6 shows the class-wise performance. The accuracy of the tail class is much higher than that of the head class. The performance degradation on head classes may be caused by the false positive predictions on head classes, i.e., the standard training method tends to classify tail classes as head classes, resulting in high accuracy on the head classes. The bad performance on tail classes when using standard training also proves this. In addition, the overall accuracy and F1-score show that MARR alleviates the unrectified prediction problem.

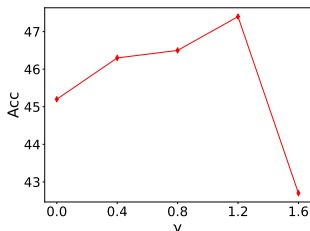

Figure 4: Accuracy on CIFAR-100-LT(200) with different $\gamma$.

**Effects of different $\gamma$**   To explore the effect of different $\gamma$, we also conduct experiments and visualize the performances on all CIFAR-100-LT(200). The results are shown in Figure 4. We can observe that compared 1.2 is the best compared with other values. $\gamma$ can not be too large since in this way the weight for head

classes is too small. It is worth noting that MARR also achieves 45.2% accuracy when $\gamma$ is 0. This means MARR still works even if we do not use any loss re-weighting techniques in the second stage. For other datasets, we directly use 1.2 for $\gamma$.

## 5 Conclusions

This paper studied the long-tailed visual recognition problem. Specifically, we found that head classes tend to have much larger margins and logits than tail classes. Motivated by our findings, we proposed a margin rectification function with only $2K$ learnable parameters to obtain the balanced logits in long-tailed visual recognition. Even though our method is very simple to implement, extensive experiments show that MARR achieves favorable results compared with previous methods without altering the model representation. We hope that our study on logits and margins could provide experience for joint optimization of the model representation and margin rectification.

In the future, we aim to develop a unified theory to better support our algorithm design and apply this algorithm to more long-tailed applications.

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
