# OpenReview forum: "Margin Rectification for Long-Tailed Visual Recognition"
_TMLR — Rejected by TMLR_

### Review · Reviewer_dArX · 2022-04-23

**Summary Of Contributions:**

This work studies long-tailed visual recognition from a margin-based perspective: 1) This work observes a typical classifier trained on a long-tailed dataset has larger margins and logit values for head classes (which have more training samples) than tail classes (which have less training samples). 2) Motivated by this observation, this work proposes a simple-yet-effective MARgin Calibration (MARC) approach to balance the margins and logits for different classes. Specifically, after the standard training of a neural recognition model, MARC fixes the feature representation and linearly transforms the logits to calibrate the final classification layer. MARC is easy to implement, and experiments on multiple benchmarks demonstrate its effectiveness.

**Broader Impact Concerns:**

The reviewer does not have ethical concerns for this work.

**Requested Changes:**

1. Experiment concerns:

- The proposed margin calibration method MARC is coupled with the loss re-weighting technique (Eq. 12, originally from [1][2]). Although Fig. 6 has shown some results of the effect of this component, it is not very clear how MARC solely improves the long-tailed recognition performance, especially on larger datasets.
- As mentioned above (see “Weaknesses 2”), one may merge the new parameters into the existing linear classification layer. If in the calibration stage, the linear classification layer is properly re-initialized (considering ||W_j|| in Eq. 9), and trained with the re-weighted loss, would the final model be equivalent?

2. More clarification required:
- As mentioned above (see “Weaknesses 2”), the connection between the motivation and method is not strong in the submission. It would be great if this connection can be better explained.
- As mentioned above (see “Weaknesses 3”), the improvement over previous work on decision boundary adjustment [1][3] seems insufficient. It would be great if this work can clarify the knowledge advancement over previous approaches.
- In Sec. 3.2, the average margins/logits are computed over only samples belonging to the same class j. However, when applying MARC, the calibration transformation is applied to all samples (of course, the true label of each data point is unknown to the model). It would be great if this difference can be clarified.

3. Minor issues:
- The terms “logits” and “classification scores” are interchangeably used in this work, referring to the linear classification layer outputs before Softmax. Some other literature may consider the outputs after Softmax, normalized to probability distributions, as “classification scores”. It would be more consistent if this work keeps using “logits”.

[1] Songyang Zhang, Zeming Li, Shipeng Yan, Xuming He, and Jian Sun. Distribution alignment: A unified framework for long-tail visual recognition. In CVPR, 2021.

[2] Yin Cui, Menglin Jia, Tsung-Yi Lin, Yang Song, and Serge Belongie. Class-balanced loss based on effective number of samples. In CVPR, 2019.

[3] Bingyi Kang, Saining Xie, Marcus Rohrbach, Zhicheng Yan, Albert Gordo, Jiashi Feng, and Yannis Kalantidis. Decoupling representation and classifier for long-tailed recognition. In ICLR, 2019.

**Strengths And Weaknesses:**

Strengths:
1. The proposed method, MARC, is easy to implement and effective. It could serve as a good baseline for future research of long-tailed recognition.
2. This work starts from a perspective of margin imbalance, which has not been well-studied in the previous literature. This perspective could potentially inspire novel designs for loss function engineering or decision boundary adjustment approaches.
3. Comparison on various benchmarks effectively validates the proposed method.
4. The writing is clear and easy to follow.

Weaknesses:
1. This work is based solely on empirical observations and experiments. A solid theoretical analysis from the margin perspective is missing from the submission.
2. The connection between the motivation and the proposed method seems not well-established in this work. Although this work is motivated by the observation of imbalanced margins and logits, the proposed method does not explicitly guarantee balanced margins and logits after calibration. Experiment results indeed show more balanced margins and logits after applying MARC (Fig. 1, 3, 4), but it is not well-explained how this balance is achieved.
3. The technical advantages of this work over DisAlign[1] is unclear: Mathematically, the additional parameters (omega and beta in Eq. 9) can be absorbed into the linear classifier layer, since Eq. 9 is just another linear transformation on the output. This class-specific linear transform can be considered as one part of the adaptive calibration function in [1] (see Eq. 4 in [1]), except that the weight magnitude (||W_j|| in Eq. 9) is explicitly considered in the initialization. Also, from experiments on larger datasets (Tables 3, 4, 5), the performance of the proposed method MARC is very close to DisAlign.
4. From Tables 3&4, the proposed method is sacrificing its performance on many-shot classes for few-shot classes and overall performance. Compared to previous methods, the performance drop on many-shot is more significant, which is not ideal if accurately recognizing common categories should be guaranteed first.

---

### Review · Reviewer_rgQZ · 2022-04-23

**Summary Of Contributions:**

This work aims to improve long-tailed accuracy for visual recognition by balancing the margins of a pre-trained classifier.
The adjustment for balancing is parameterized as a scale and shift of the class-wise logits.
This approach of adjusting the classification margins is justified by experiments showing that the magnitude of logits and hence margins decreases from head to tail classes.
Intervening after classifier training to calibrate the margins does indeed improve overall accuracy.
The method for margin calibration, MARC, optimizes affine parameters to adjust the predicted logits (Eq. 8) while reweighting the loss by inverse frequency (Eq. 12).
These parameters, 2K in total for K classes, are trained after the classifier training on the same training data.
This brief description of the method underlines its simplicity.
There is a slight difference with prior methods, in that the existing Decouple-LWS method optimizes weight norms alone, which is equivalent to scaling margins, while this work scales and shifts margins.
MARC improves accuracy across multiple long-tail benchmarks for visual recognition, in particular the long-tail editions of CIFAR, ImageNet, Places, and iNaturalist.
In many several cases MARC achieves higher accuracy, and on overall accuracy it is the best or competitive with the best (<1 point difference).
However, this improvement is not universal, and MARC does significantly worse (5+ points absolute) than one or two baselines on head (common) or tail (rare) classes.

Claims:

1. The first analysis of margins for calibrating long-tailed predictions.
2. A simple method for margin calibration, MARC, with only 2K parameters, and few lines of code required for implementation.
3. Good empirical accuracy on standard benchmarks.

**Broader Impact Concerns:**

This work does not raise ethical concerns in its data or goals. The datasets
chosen are standard benchmarks that do not include sensitive content:
CIFAR-10/100 and ImageNet-1k consist of generic object classes without human
data while places consists of scene classes from internet data, and iNaturalist
consists of various plant, animal, and microbe classes from naturalist citizien
science data. The purpose of this work is long-tailed recognition, which seeks
to improve recognition across common and rare classes, and as such could serve
to increase fairness by countering increases in error for rarer data that may
come from minorities in demographic, geographic, economic, or other aspects.

**Requested Changes:**

**Critical Changes**

- Please sharpen the third claim about empirical performance (Introduction). Is MARC state-of-the-art? If not, does it outperform certain classes of methods but not others? Where is it only competitive, or less than competitive? The empirical claim should give some sense of this, to orient the audience to the results tables. The current text of "performs notably well" is too vague.
- Please coment on the use of "calibration" in the title and method name (Title, Introduction). Probabilistic calibration is already a defined term (see "On Calibration of Modern Neural Networks" by Guo et al. ICML'17 for reference), where in this work the purpose seems to be more about balance among head and tail classes. Of course calibration can also have this general sense of equalization, but clarification could be helpful, by explaining in the introduction that the topic is not calibration in the formal probabilistic sense.
- Please provide a reference for margin in Sec. 3.2, to make clear that it is a well-established concept and not an invention of this paper. Consider a statistical learning textbook like "The Elements of Statistical Learning" or the work of Vapnik.
- Please clarify what "harm" and "good" are for a representation (Sec. 3.2). Is it simply a matter of increasing and decreasing accuracy of the deep network classifier? Is there a measure of the representation itself, apart from task accuracy, that is changed by these techniques?

**Suggested Improvements**

- Figure 1 (a): What is the y axis exactly? Is it the norm of the logits, or the magnitude of the top logit? Please specify. To emphasize the point about balancing logits and margins, the caption could describe the dispersion of logits/margins before/after MARC, by measuring the range or the variance.
- Method explanation: How is the adjustment of the margins "dynamic"? ("to dynamically adjust the initially learned margins"). Dynamic has the sound of input-conditional or test-time updates to inference, but the trained scaling and shifting of the logits is static and shared across all teset inputs. Consider dropping "dynamic" from the description.
- Preliminaries: Is it necessary for the test data to be balanced? It seems natural and practical for long-tailed problems to be long-tailed at both training and testing time, provided there is an appropriate metric such as average precision (AP) to account for the disparity in frequency.
- Ablation: The sweep over gamma should evaluate on ImageNet-LT as well as CIFAR to confirm its findings. This could be visualized alongside the CIFAR result.

**Trivial Feedback**

- [clarity] Define head and tail classes in the abstract, by for instance parenthetically specifying "head (common)" and "tail (rare)" classes to reach audiences outside of the niche of long-tailed recognition.
- [clarity] Contrast the method with other work in the conclusion to better highlight next steps. For instance, MARC is able to improve long-tailed recognition without altering the model representation, but future research could explore joint optimization of the model representation and margin calibration.
- [clarity] In Figure 1 (c), consider removing the line plot that is overlaid on the bar chart, as it interferes with reading the bars.
- [wording] Replace "indexes" with "indices".
- [wording] Replace "technologies" with "techniques" in the "Data re-sampling" paragraph of the related work.
- [wording] Replace "neuron networks design" with "architecture design".
- [wording] Replace "It is well noting" with "It is worth noting".
- [grammar] Delete initial "the" in the abstract to start with "Long-tailed visual recognition tasks".
- [grammar] Delete "the" in "In addition, the recent research".
- [grammar] Replace "decision boundary adjusts methods" with "decision boundary adjustment methods".
- [grammar] Capitalize "the" in "the relation between the uncalibrated".



**Strengths And Weaknesses:**

**Strengths**

- Sheer simplicity: The method adjusts inference with minimal code and computation by an affine transformation of the logits. The transformation is shared across all data, that is, the parameters are constant. These parameters are optimized in a secondary phase of training, with the classifier parameters fixed, on the same training data.
- Parameter efficiency: The method only requires 2K parameters for K classes. This is a marginal increase given that many tasks have only 100s-1000s of classes while deep nets can have many layers and millions of parameters or more. These parameters are merely multiplicative and additive constants for scaling and shifting predicted logits.
- Comprehensive datasets and baselines: the long-tail editions of CIFAR-10/100, ImageNet, and Places are standard choices for evaluation. The baselines evaluated are sufficiently strong, recent, and diverse as many were published in the last 2-3 years and they span methods that alter sampling, losses, and decision boundaries among others. See Table 2 for instance as an example of the baselines covered.
- Self-containment: The paper stands on its own, by explaining vocabulary and recapitulating derivations like the definition of margin. Method details and experiment settings are adequately explained, and the method in particular is described by text, pseudo-code, and lines of pytorch code.
- Analysis: Plots of logits and margins show that the range of each is reduced by MARC, and this pattern is less visually obvious for the instance-wise decision boundary adjustment DisAlign.
- Ablation: The main hyperparameter gamma is swept on CIFAR-100-LT to show that (1) the method can work without loss re-weighting, (2) excessively high values hurt accuracy by under-weighting head classes, and  (3) the chosen value of 1.2 is best.

**Weaknesses**

- There are negative experimental results for accuracy. MARC harms accuracy on head/common classes relative to the standard softmax on ImageNet and Places. While it improves accuracy on classes of medium and few instance counts over the softmax, it is outdone or rivaled by existing decision boundary adjustment methods (see Tables 3, 4, 5).
- The technical difference with decoupled decision boundary adjustment methods, in particular Decouple-LWS, is quite slight: instead of rescaling class-wise weight norms the logits are both scaled and shifted with class-wise affine transformation parameters. Nevertheless it is different, and the results are slightly different.
- Results for each dataset are only shown for a single classifier per dataset. However the chosen classifiers are good, common choices (ResNets, with the exception of one ResNeXt, which is likewise an established architecture). Nevertheless, the results would be more convincing if the effect of margin calibration were shown to be consistent across multiple models, by perhaps evaluating MARC across different models on ImageNet-LT, such that the computational burden is not too high.
- The comparison with DisAlign disagrees with the results of the original paper, but the difference results from controlling for standardization of pre-training so this is only a minor issue.

In summary, the claims of this work are mostly supported and the findings are of interest to researchers on long-tailed visual recognition.
Where the claims are lacking it is not so much for lack of evidence as for vagueness of description.
The interest to the audience is principally in the improved empirical performance and the simplicity of the implementation, which should permit straightforward reproduction and extension to different data, improved methods, and so forth.
See the requested changes for more concrete comments and actional feedback on these points.

---

### Review · Reviewer_6jWb · 2022-04-23

**Summary Of Contributions:**

The papers studies the margins and logits for multiclass classification, and proposes a method to improve long-tailed classification on image datasets. The authors show that the margin and logit values are smaller for tail classes, and call these "uncalibrated". The authors propose "MARC", an algorithm to recalibrate the model which outperforms other methods on long-tail versions of standard benchmarks (ImageNet, CIFAR, Places, INat).

MARC consists of a learnable affine transformation of the logits (logits' = a*logits + b*weight_norm). The parameters are learned in a second phase of SGD with the network parameter fixed.



**Broader Impact Concerns:**

The paper proposes a fundamental algorithm for recalibration and evaluates on standard datasets. No concerns.

**Requested Changes:**

Please address the major weaknesses above, in particular, the relationship to Platt scaling, and framing of low-confidence logits as "uncalibrated".

**Strengths And Weaknesses:**

Strengths

- The method is simple.
- There are multiple evaluation sets, and performance looks good compared to other methods.
- The paper is quite clearly written.

Weakness (major)

- As far as I can tell MARC is equivalent to Platt scaling [1]. (See [2] for usage in modern neural nets). The only difference is that the offset term is re-parameterized as the offset/weight_norm_for_corresponding_class. For the paper to be of interest, it would be great if the authors could clarify how this method relates to, or differs from Platt Scaling, and have an empirical comparison.

- The paper frequently refers to "uncalibrated logits" or "uncalibrated margins". These phrases do to not fully make sense to me. Calibration is a property of a probabilistic prediction; and a low-confidence prediction (i.e. one with a small margin) may be perfectly calibrated (if the mistake rate equals the predict confidence).

- The literature review discusses methods for long-tailed correction, but not many standard recalibration methods (e.g. those in [2]).

- Figure 1 presents the correlation between the logit magnitude and the margin as an interesting empirical finding. However, this seems somewhat trivial to me (unless I am misunderstanding something?): unless the weight norm for the class weight vector is inversely correlated with the average logit magnitude for the class (which would be surprising), this correlation will always be observed.

Weaknesses (minor)

- The authors reuse the training set for recalibration. I think this should be discussed, because it is usual to use a hold-out set for recalibration.

- MARC contains two additional (known) tricks --- balanced sampling and loss weighting. The latter is ablated in Fig 6, the former I feel should also be ablated (to show that a benefit comes from the claimed contribution of MARC).

- A few typos, a couple I saw were:

"decision boundary adjusts methods" -> "decision boundary adjustment methods"

"It is well noting" -> "It is worth noting"


[1] http://citeseer.ist.psu.edu/viewdoc/summary?doi=10.1.1.41.1639
[2] https://arxiv.org/abs/1706.04599

---

### Author Response · Authors · 2022-05-07
**New revision**

Dear ACs and reviewers,

Thank you for your constructive feedback! Now we upload a new revision of the paper according to your comments, which mainly contains the following modifications:

1. We renamed the method from "calibration" to "rectification", which is more concise according to the comments of most reviewers. Thus, the main method name will be MARR, not MARC. (R for rectification)
2. We thoroughly re-examined all the minor weaknesses mentioned by reviewers and modified them.
3. We added more related work to discuss our contribution.
4. We resolved many questions and concerns raised by reviewers.

We hope that you can like it. If you have any questions, please reply to us:)

---

> ### Comment · Action_Editors · 2022-05-07
> **Please update your manuscript**
>
> Thanks for submitting to TMLR. As the general guidelines of TMLR, the authours may have the chance of updating the manuscript accordingly (https://jmlr.org/tmlr/ae-guide.html). So please also update your manuscript as commented. Please using different color to highlight the changes of the content in the pdf.

---

> > ### Author Response · Authors · 2022-05-07
> > **Changes marked with blue color**
> >
> > Thanks for the kind reminder! Now we uploaded a new version with changes marked with blue color:)

---

### Decision · Action_Editors · 2022-05-22

**Recommendation:** Reject

**Comment:**

Thanks for submitting to TMLR. This paper is reviewed by three very professional researchers who have done many excellent works on the related topics. The reviewers give very details and insightful suggestions, and comments. Hopefully these comments can help authours to significantly improve the quality of this paper.

There are several major points that make all the reviewers believe that this paper unfortunately can not be accepted unless a very thorough revision. Particularly,
(1) "As far as I’m concerned, the only difference between MARC and DisAlign is how to initialize the learnable affine transformation."
(2) "The biggest concern is still the difference between the proposed MARR and the previous method Platt scaling. Furthermore,  The empirical performance advantage of MARC is unclear."
(3) "there is well-established research on calibration and decision boundary scaling (like Platt scaling), and so this perspective is not entirely different. Given this context, the contribution of this work is more to re-confirm the use of such approaches for long-tailed recognition in particular."

Thus it is very advisable to revise, reorganize the paper, and possibly add more empirically validation to support the claims in the future version. Thanks the great efforts to all reviewers, and thanks again for submitting to TMLR.